# ZeroBind: a protein-specific zero-shot predictor with subgraph matching for drug-target interactions

Yuxuan Wang [1], Ying Xia [1], Junchi Yan [2], Ye Yuan [1], Hong-Bin Shen [1] & Xiaoyong Pan [1] ✉

Existing drug-target interaction (DTI) prediction methods generally fail to generalize well to novel (unseen) proteins and drugs. In this study, we propose a protein-specific meta-learning framework ZeroBind with subgraph matching for predicting protein-drug interactions from their structures. During the meta-training process, ZeroBind formulates training a protein-specific model, which is also considered a learning task, and each task uses graph neural networks (GNNs) to learn the protein graph embedding and the molecular graph embedding. Inspired by the fact that molecules bind to a binding pocket in proteins instead of the whole protein, ZeroBind introduces a weakly supervised subgraph information bottleneck (SIB) module to recognize the maximally informative and compressive subgraphs in protein graphs as potential binding pockets. In addition, ZeroBind trains the models of individual proteins as multiple tasks, whose importance is automatically learned with a task adaptive self-attention module to make final predictions. The results show that ZeroBind achieves superior performance on DTI prediction over existing methods, especially for those unseen proteins and drugs, and performs well after fine-tuning for those proteins or drugs with a few known binding partners.

Identifying the interactions between drugs and targets (proteins) plays a crucial role in the process of drug discovery[1–3]. However, the traditional experimental methods for resolving the crystal structures of drug-protein complexes to identify the drug-target interactions are costly and time-consuming[3–5]. In order to reduce costs, in silico approaches are gaining more attention. Instead of taking massive candidates into an in vitro search, it is more efficient and less costly to use computational approaches to virtually screen out most candidates prior to an in vitro search. Generally, there are two major groups of in silico approaches, docking simulations and data-driven learning-based methods. Docking simulations utilize the 3D structure of drug molecules and target proteins to identify their potential binding sites, which are also time-consuming[1,6,7]. In contrast, due to the rapid development of machine learning, utilizing the features derived from proteins and drugs to identify their interactions achieves both high accuracy and low cost.

Data-driven learning methods generally formulate drug-target interaction (DTI) prediction as binary classification or regression tasks[8–11], where interacting pairs of proteins and drugs are extracted from existing databases like BindingDB[12], CHEMBL[13], PDBbind[14,15], and DrugBank[16]. Since the nature of potency values is logarithmic, a decrease in kinetic constants from micromolar to nanomolar levels results in an exponential change. Thus, the output values of regression tasks are normally defined as the negative log of kinetic constants, i.e., $K_i$, $K_d$, $IC_{50}$ and $EC_{50}$. For classification tasks, a threshold is set to define binding or nonbinding according to $K_i$, $K_d$, $IC_{50}$, and $EC_{50}$[2,17,18].

[1]Institute of Image Processing and Pattern Recognition, Shanghai Jiao Tong University, and Key Laboratory of System Control and Information Processing, Ministry of Education of China, Shanghai 200240, China. [2]Department of Computer Science and Engineering, and MoE Key Lab of Artificial Intelligence, AI Institute, Shanghai Jiao Tong Univenisty, Shanghai 200240, China. ✉e-mail: 2008xypan@sjtu.edu.cn

Machine learning-based methods from molecules and proteins features mainly focus on learning a good representation of molecules and proteins, which are then fed into a classification/regression model to perform the prediction task[8-11,19].

Recently, deep learning has achieved exciting results in DTI prediction by learning from known drug-target interactions. However, these methods cannot generalize well to those unseen proteins and drugs[8,20]. Similarity/distance-based[20-23] and network-based[24-26] methods, which utilize protein-protein similarity, drug-drug similarity and known DTIs, achieve overestimated performance on the transductive test[9], where both proteins and ligands in the test set are present in the training set. In addition, there are often few known binding drugs with a newly discovered target protein, making it difficult to work on the inductive test, where both proteins and ligands in the test set are absent in the training set. Currently, most existing methods focus on how to effectively learn representations of molecules and proteins, and then feed the representations into classification or regression models[8-11,19]. The representation learning methods, like DeepPurpose[8], take molecular fingerprints and SMILES[27] strings as the molecular features and amino acid sequences as protein features. Then they apply a convolutional neural network (CNN)[28], recurrent neural network (RNN)[29] or Transformer[30] model to embed the input features. To take spatial information into consideration, some methods apply CNNs to 3D images derived from structures of proteins and molecule[31]. GNN[32] has gained much attention in recent years due to its capability of handling graphs as non-Euclidean structured data[33]. Other methods utilize GNNs to embed graphs of proteins and molecules to perform DTI prediction[10]. DTI prediction also benefits from the rapid development of GNNs since both drug and target protein are naturally graph-structured data. GEFA[34] combines pre-trained protein embeddings and a graph-in-graph neural network with an attention mechanism to capture the interactions between drugs and protein residues. Thus, it is reasonable to use GNNs for the representation learning of proteins and drugs.

However, classical pair-input approaches often take drug-protein pairs as training samples by concatenating and feeding their representations into a dense layer to identify the interactions. However, these approaches may fail at inductive tests for unseen proteins and drugs not seen during the model training, which is due to their potential shortcuts[35] that remember the degree ratio of binding annotations in the training set rather than learning the molecular features of interactions[9]. To generalize to unseen proteins and drugs, AI-Bind leverages network-derived negatives and pre-training to resolve the shortcut learning issue[9]. Furthermore, proteins may have diverse binding patterns with drugs, which can be captured by training a protein-specific model for individual proteins. Recently, a ligand-based method MetaDTA[36] is developed as a few-shot predictor for proteins with a few known binding drugs under the meta-learning framework. However, MetaDTA does not support zero-shot prediction for unseen proteins without known binding drugs in the training set.

To address these challenges, we propose a protein-specific zero-shot predictor ZeroBind for drug-target interaction prediction under the meta-learning framework. ZeroBind adopts the meta-learning framework MAML++[37] as the training strategy and each base model makes binding drug predictions of a specific protein, which mainly consists of four modules: (1) a Graph Convolutional Network (GCN) encoder learning the embeddings of the molecule graph and protein graph, (2) a Subgraph Information Bottleneck (SIB) module generating the essential IB-subgraph of the protein graph as a potential binding pocket, (3) a Multilayer Perceptron (MLP) module concatenating the protein IB-subgraph embedding and molecular embedding to perform DTI prediction, and (4) a task adaptive self-attention module to measure the importance of different tasks, where different DTI tasks contribute differently to the meta-learner. The generated task weight is used for weighted average loss and further incorporated into the meta-learning procedure.

In summary, our contributions are as follows: (1) We formulate the drug-target interaction prediction for unseen drugs and proteins as a zero-shot learning problem. The general knowledge of DTI prediction learned from existing proteins, drugs, and their interactions with meta-learning strategy have a greater ability for generalization to unseen proteins and drugs than existing methods. (2) We train one DTI task model per protein, where task adaptive self-attention is designed to calculate the contributions of multiple DTI tasks to the protein-specific meta-learner. Thus, each protein-specific model captures individual binding patterns to drugs. (3) We propose model-agnostic IB-subgraph learning to automatically discover compressed subgraphs as potential binding pockets in proteins instead of redundant graph information derived from the whole protein. (4) We conduct extensive experiments on three independent zero-shot test sets and one few-shot test set. Results show that ZeroBind consistently outperforms existing methods. Further validation of real-world SARS-COV-2 drug-target binding prediction demonstrates the reliability of ZeroBind predictions and the subgraphs detected by IB-subgraph learning align well with the known binding pockets in proteins.

## Results
### Overview of ZeroBind
In this study, ZeroBind formulates the DTI prediction as a meta-learning task and proposes a meta-learning framework to solve the generalization problem of unseen proteins and drugs in DTI prediction. Specifically, a meta-learning task is defined as binding drug predictions of a specific protein, where IB-subgraph learning is leveraged to automatically discover compressed subgraphs as potential binding pockets in proteins and a self-attention mechanism is designed to learn weights for each task of a protein. The flowchart of ZeroBind is illustrated in Fig. 1.

In detail, ZeroBind uses network-based negative sampling[9] as data augmentation to alleviate the annotation imbalance (Fig. 1a, Methods section). Figure 1b, c illustrates the ratio of positive samples before and after network-based negative sampling on the training set, indicating that network-based negative sampling alleviates the annotation imbalance to a certain extent. Then, it samples the DTIs into the support and query set (Fig. 1d), where the support set is used to train the meta-learner and the query set is used to train the task-specific models. After repeating N inner steps, all losses are weighted to optimize the meta-learner with gradient descent. For each protein, ZeroBind trains a DTI prediction task. Figure 1e gives the architecture of the base model in ZeroBind, where the protein graph and molecule graph are fed into a backbone GCN to learn embeddings for drugs and proteins. Furthermore, a weakly supervised Subgraph Information Bottleneck (SIB) module is designed to model and discover potential binding pockets in proteins. The SIB module not only reduces redundant information to boost the performance, but also brings interpretable insights into ZeroBind by identifying the critical residues in the protein. Figure 1f introduces an adaptive self-attention module to measure the contribution of each task of a protein, where different DTI tasks contribute differently to the meta-learner. ZeroBind support predicting DTIs in zero-shot and few-shot scenarios. The former uses the meta-learner to make predictions directly without fine-tuning using the samples of proteins in meta-testing, and the latter uses the protein-specific model to make predictions with fine-tuning using the samples of the protein in meta-testing.

### ZeroBind outperforms existing methods in both zero-shot and few-shot settings for DTI prediction
To demonstrate the advantages of ZeroBind, we compare it with multiple baseline methods and calculate the area under the receiver operating characteristic (AUROC) and the area under the precision-

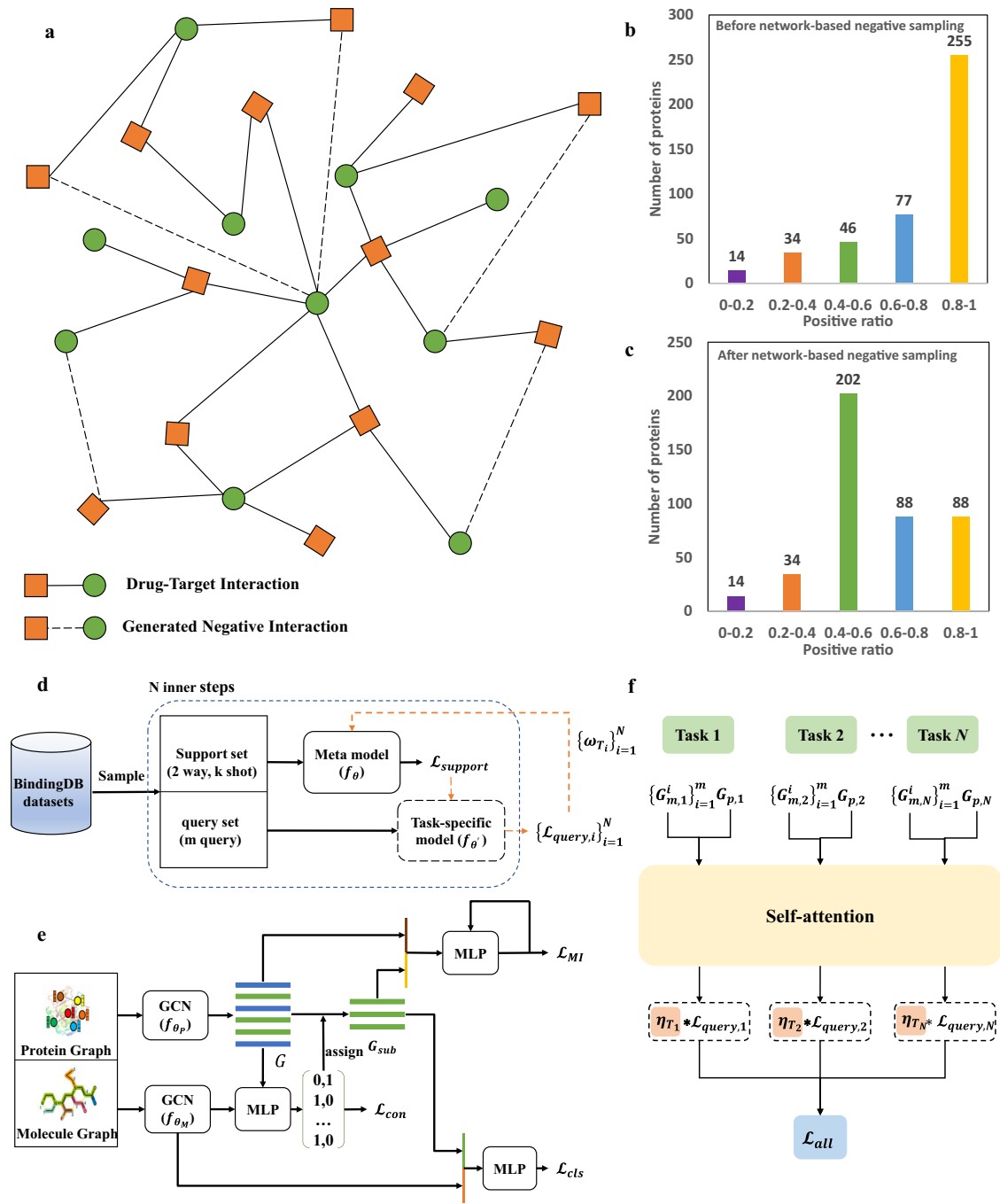

**Fig. 1 | The framework of ZeroBind. a** Network-based negative sampling strategy. The bipartite network consisting of drugs and protein targets: The square nodes represent the protein nodes and the circle nodes represent the molecule nodes, and there are only edges between different types of nodes, representing the corresponding drug-target interaction. Solid lines represent existing drug-target interactions (DTIs) and dotted lines represent the generated negative interactions with the shortest path distance $\geq 7$. **b** The positive ratio of the training set before the network-based negative sampling strategy. **c** The positive ratio of the training set after the network-based negative sampling strategy. **d** Given the support set and query set, $\mathcal{L}_{support}$ is first calculated and utilized to update the base model with parameter $\boldsymbol{\theta}$ to a task-specific model with parameter $\boldsymbol{\theta}'$ using the support set of each task, and then the task-specific model calculates the $\mathcal{L}_{query}$ using the query set of the task. After repeating N inner steps, all losses are weighted average by $\{\boldsymbol{\omega}_{T_i}\}_{i=1}^{N}$, and gradient descent is further performed to optimize the meta model.

**e** The architecture of the base model in ZeroBind. For each task, the protein graph and the molecule graph are fed into a backbone graph convolutional network (GCN) with parameters $\boldsymbol{\theta}_{\mathbf{P}}$ and $\boldsymbol{\theta}_{\mathbf{M}}$, respectively, to obtain their embeddings. Subsequently, a SIB module is proposed to generate the IB-subgraph of a protein as potential binding pockets in a weakly supervised way. The protein subgraph embedding is concatenated with the molecular embedding and they are fed into a Multilayer Perceptron (MLP) module to identify the interactions. **f** Task adaptive attention module. It takes the concatenation of the protein embedding $\mathbf{G}_{\mathbf{p,k}}$ and the average of all molecule embeddings $\{\mathbf{G}_{\mathbf{m,k}}^{\mathbf{i}}\}_{\mathbf{i=1}}^{\mathbf{m}}$ in the query set as the task embedding. After using the self-attention layer to compute the weight of each task, denoted as $\{\boldsymbol{\eta}_{T_i}\}_{i=1}^{N}$, the overall loss is averaged and incorporated into the meta-training process for updating the model parameters. Source data are provided as a Source Data file.

recall curve (AUPRC) on three independent test sets and one few-shot test set. To ensure the effectiveness of performance comparison, we conduct five independent experiments with five random seeds for datasets partitioning and model training, and report the average results along with standard deviation.

The overall performances of all methods in three independent test sets are reported in Fig. 2a (Supplementary Table 1). ZeroBind achieves an AUROC of 0.9521(±0.0034), 0.8681(±0.0052), 0.8139(±0.0035) in the Transductive, Semi-inductive, and Inductive test sets, respectively. We can see that ZeroBind outperforms all baseline methods in the Transductive, Semi-inductive, and Inductive test sets. Compared to the best baseline method, ZeroBind achieves relative improvements in AUROC with 2.86% on the Transductive test set, 10.29% on the Semi-inductive test set, and 3.38% on the Inductive test set, the corresponding t-test p-value is $1.02 \times 10^{-6}$, $8.02 \times 10^{-11}$, $3.06 \times 10^{-7}$, respectively. Moreover, the relative improvements of AUPRC are 1.00% on the Transductive test set, 0.96% on the Semi-inductive test set, and 1.21% on the Inductive test set compared to the best baseline method, the corresponding t-test p-value is $8.93 \times 10^{-8}$, $5.15 \times 10^{-3}$, $1.56 \times 10^{-5}$, respectively. In addition, on the three test sets, we observe that the performance of all methods decreases, indicating that a certain external shortcut learning exists for the Transductive test set.

Figure 2a shows that AI-bind and our proposed ZeroBind outperform other baseline methods on the Inductive test set due to the incorporation of network-based negative sampling and unsupervised pre-trained embeddings to avoid the potential shortcut learning. ZeroBind achieves stable and better performance at Transductive and semi-inductive test sets than baseline methods, indicating that ZeroBind can effectively learn useful embeddings of proteins and drugs. Furthermore, ZeroBind achieves better performance in the inductive test set than AI-Bind. The potential reason is that ZeroBind uses a meta-learning framework to obtain a general knowledge of DTI prediction across multiple proteins, which can generalize well to the DTI prediction of unseen proteins and drugs.

We also evaluate the protein-specific DTI prediction performance of ZeroBind with other baseline methods on the inductive test set and semi-inductive test set (Fig. 2b). After excluding proteins without binding or nonbinding labels, we obtain 775 proteins with binding

drugs and one task model is trained per protein, here ZeroBind is evaluated on the combined inductive and semi-inductive test sets. ZeroBind achieves an average AUROC of 0.78, which is higher than the AUROC 0.66 of DeepConv-DTI, 0.68 of GraphDTA, 0.69 of Deeppurpose, 0.75 of AI-bind and 0.73 of DrugBAN across the 775 proteins. Of the 775 proteins, ZeroBind outperforms DeepConv-DTI, GraphDTA, Deeppurpose, AI-bind, and DrugBAN for 525, 491, 305, 180 and 346 proteins, respectively. For each method, we present the number of proteins that this method outperforms other methods (Fig. 2c) according to the number of binding molecules. We observe that ZeroBind outperforms other baseline methods for the most proteins in the three ranges, especially for the proteins with only 1–10 known binding molecules, followed by AI-Bind.

We further evaluate ZeroBind against the baseline methods on the few-shot test set. ZeroBind and other baseline methods are first trained with the training set as the pre-training model, and further fine-tuned on each few-shot fine-tuning set and evaluated on the corresponding few-shot test set. The performance of ZeroBind model and other baselines on the few-shot test set is shown in Fig. 2d (Supplementary Table 2). In the few-shot test, ZeroBind outperforms the best baseline methods with relative improvements of AUROC 1.55% and AUPRC 1.62%. The potential reason is that the meta-learning framework has the strong generalization ability to quickly adapt to additional protein tasks with a few training samples. We can see that fine-tuning improves the zero-shot prediction performance by a large margin, which is very useful for the real-world application scenario that a protein has only a few known binding drugs.

## ZeroBind detects the subgraphs that align well with known binding pockets of proteins in a weakly supervised way

ZeroBind does not use binding pocket information as ground truth labels for model training, instead, it uses the global DTI labels as a weakly supervised labels. To demonstrate that the IB-subgraph module in ZeroBind is able to detect the binding pocket in the protein, we use the Jaccard similarity coefficient[38] to compare the predicted binding pocket against the true binding pocket on PDBbind dataset[14,15]. The true binding pocket information containing pocket residues and 3D coordinate locations are downloaded from PDBbind, which contains 14,000 binding residues for 535 proteins. The Jaccard similarity

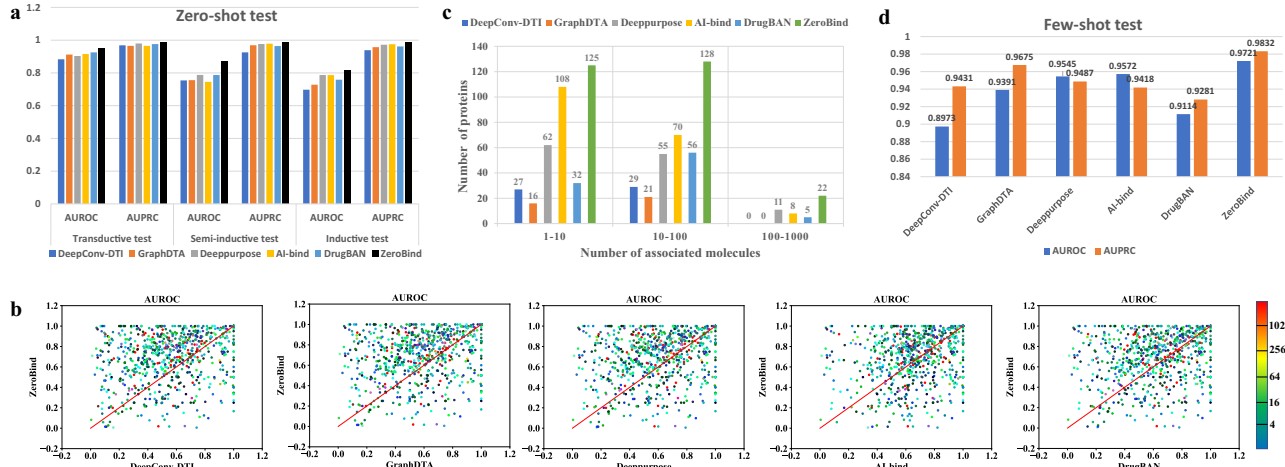

**Fig. 2 | Performance comparison of ZeroBind with baseline methods in zero-shot and few-shot scenarios. a** Zero-shot performance evaluation with ZeroBind and baseline methods on three independent test sets. **b** Area Under the Receiver Operating Characteristic Curve (AUROC) comparison of the protein-specific ZeroBind and the baseline methods for 775 proteins in the combined inductive and semi-inductive test set. The color of points stands for the number of training proteins. **c** The number of proteins that the method performs the best among the

compared methods for the combined inductive and semi-inductive test sets. **d** Few-shot performance comparison of ZeroBind with baseline methods on the few-shot test sets. Supplementary Tables 1 and 2 provide the data statistics for zero-shot and few-shot prediction with two-sided t-test without adjustment. Source data are provided as a Source Data file. AUPRC refers to Area Under the Precision-recall Curve.

coefficient is used to calculate the degree of intersection between two sets of individual DTIs, and the formula is defined as *Jaccard similarity coefficient*$(\hat{P}, P) = \frac{\hat{P} \cap P}{\hat{P} \cup P}$, $\hat{P}$ represents the set of predicted binding pocket residues by ZeroBind for a DTI, and $P$ represents the set of true binding residues in the pocket for this DTI.

In addition, we calculate the Jaccard similarity coefficient of predicted binding pockets nodes and the first-order neighbors of the true binding pocket nodes as *Jaccard similarity coefficient* $(\hat{P}, P_{neighbor}) = \frac{\hat{P} \cap P_{neighbor}}{\hat{P} \cup P_{neighbor}}$, $P_{neighbor}$ represents the set of first-order neighbor pockets of the real binding pocket.

Figure 3a, b shows the distribution of Jaccard similarity coefficients of predicted binding pockets with true binding pockets and the first-order neighbors of true binding pockets, respectively. ZeroBind yields average Jaccard similarity coefficients of 0.358 and 0.605, respectively. For the neighboring pockets, Jaccard similarity coefficients are above 0.5 for most pockets. The results show that despite some discrepancy between the predicted binding pockets and true binding pockets, the predicted binding pockets are mostly around the true binding pockets, indicating the effectiveness of the generated IB-subgraph in ZeroBind as the potential binding pocket with some biological interpretability. We further conduct an experiment of randomly sampled residues as potential protein pockets, here denoted as ZeroBind^random. The results are shown in the ablation studies and here we also calculate the Jaccard similarity coefficients of randomly sampled binding residues with true

binding pockets and the first-order neighbors of true binding pockets, respectively. The ZeroBind^random yields an average Jaccard similarity coefficients of 0.013 and 0.072 (all the values are below 0.2), which is much smaller than that of ZeroBind, respectively. As shown in Fig. 3a, b, we can see that the randomly selected binding residues have little overlap with the true binding pockets or their neighbors in proteins. The results indicate that the SIB module in ZeroBind learns the potential binding pockets instead of other unrelated factors, since the DTI binding information is to a certain extent able to guide the IB-subgraph module to locate potential binding pockets.

To further demonstrate the effectiveness of the IB-subgraph module, we train a variant ZeroBind for those proteins with known binding pockets using true binding pocket as the node assignment matrix **Z** for the inductive test set. This variant ZeroBind achieves an average AUROC of 0.8278, which is higher than the AUROC 0.8032 of the ZeroBind that uses learned node assignment matrix **Z** by a narrow margin. The experimental results further validate the effectiveness of the IB-subgraph module in ZeroBind.

We further visualize the generated IB-subgraph as the potential binding pocket for Serine/threonine-protein kinase N1 protein in Fig. 3c, d. We can see that although the IB-subgraph introduces some false positive binding residues due to no known binding pocket information for IB-subgraph module training, the potential binding residues included in the IB-subgraph contain most of the true binding residues deposited in BioLip[39].

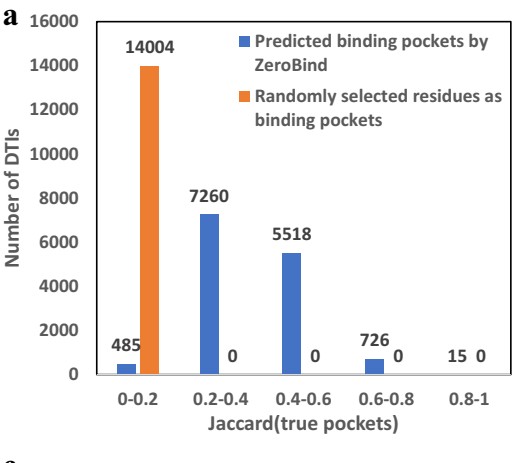
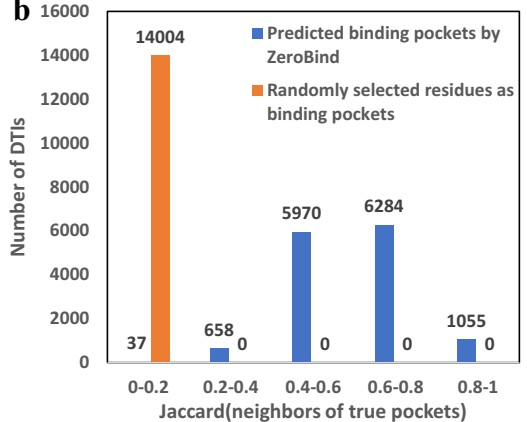
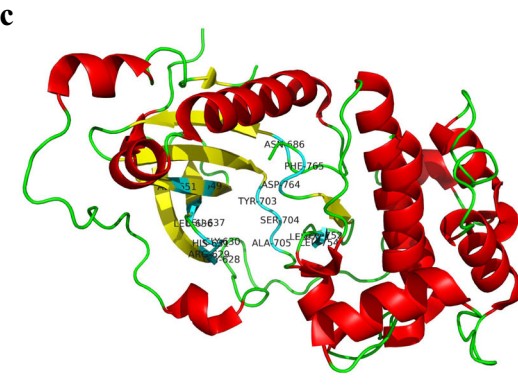
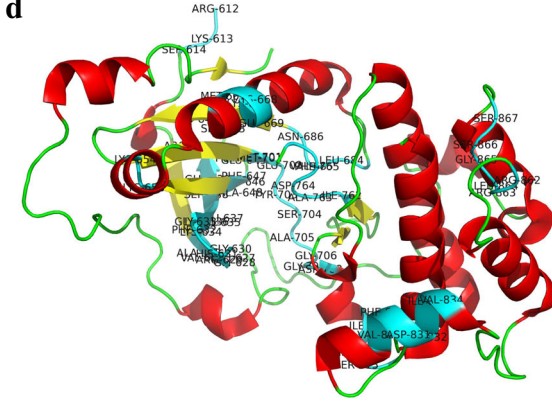

**Fig. 3 | ZeroBind is able to detect binding pockets of proteins in a weakly supervised way. a** The distribution of Jaccard similarity coefficients of the predicted binding pockets with the true binding pockets and randomly selected binding residues as binding pockets with the true binding pockets of individual DTIs. **b** The distribution of Jaccard similarity coefficients of the predicted binding pockets with the first-order neighbors of true binding pockets and randomly selected binding residues as binding pockets with the first-order neighbors of true binding pockets of individual DTIs. **c** The Serine/threonine-protein kinase N1 protein. Experimentally validated DTI binding pocket queried by BioLiP. The blue part represents the experimentally validated binding pocket along with the residue name and number. **d** The potential binding pocket predicted by IB-subgraph in ZeroBind. The red part represents the helix structure of protein, the yellow part represents the loop structure of protein and the green part represents the sheet structure of protein. Source data are provided as a Source Data file.

## ZeroBind is able to predict potential drugs against SARS-COV-2 Proteins

To better demonstrate the effectiveness of ZeroBind, we use Auto-Docking simulations to validate its predicted potential drugs targeting the SARS-COV-2 proteins. Auto-Docking simulation is a time-consuming but reliable tool to simulate the drug-target binding process. To respond quickly to public health emergencies, we need to validate ZeroBind on proteins that don't have many binding molecules. Thus, we apply ZeroBind to predict the interactions between the structures of 10 SARS-CoV-2 viral proteins not in the training set and 10,000 drugs in PubChem[40], where the PDB IDs for SARS-COV-2 protein structures are given in Supplementary Table 3. Then, we choose top-10 binding confidence pairs based on the predicted scores by ZeroBind and further perform Auto-Docking simulations to validate the effectiveness of ZeroBind predictions with AutoDock Vina[41].

The average binding affinity of the top-10 predicted pairs by ZeroBind is −7.42 kcal/mole (Fig. 4a), which are promising drug-target pairs and also validate the reliability of ZeroBind predictions[42]. As demonstrated in[9], the average binding affinities are close to −7.5 kcal/mole for three binding pairs of SARS-COV-2 proteins and drugs. Furthermore, we simulate the drug-target binding complex between ORF8 accessory protein and VZBSCWDKCMOJCR-UHFFFAOYSA-N drug in Fig. 4b, it yields a relative good binding affinity score of −8.4 kcal/mole. And we also simulate the drug-target binding complex between ORF3a protein and OLTVRSUIOUTBRQ-UHFFFAOYSA-N drug in Fig. 4c, which yields a binding affinity score of −5.8 kcal/mole. As shown in Fig. 4b, c,

we can see that the predicted drugs bind well to the pockets in SARS-COV-2 proteins. In future work, we expect ZeroBind to validate the potential drugs against the target proteins, especially those proteins with no verified drugs.

## Ablation studies on ZeroBind

To demonstrate the added value of individual modules in ZeroBind, we also conducted an ablation study to evaluate the effectiveness. The details of ZeroBind with different modules are illustrated as follows:

(1) ZeroBind$^{MAML-}$: Train the base model of ZeroBind directly without the meta-learning strategy.

(2) ZeroBind$^{SIB-}$: Using all node embeddings of the protein to identify the interaction instead of applying a SIB module to find the IB-graph on the protein graph.

(3) ZeroBind$^{attention-}$: ZeroBind without task adapted attention module to balance the importance of different tasks.

(4) ZeroBind$^{GIN}$: ZeroBind uses GIN instead of GCN as the backbone GNN.

(5) ZeroBind$^{random}$: ZeroBind sets the node assignment matrix $Z$ in Eq. (4) randomly.

The results are shown in Table 1. For ZeroBind$^{MAML-}$, we observe a significant decrease in the semi-inductive and inductive test sets, demonstrating that the meta-learning training strategy provides the powerful generalization ability to unseen proteins and drugs. We can see that ZeroBind$^{SIB-}$ yields a lower performance without the SIB

**a**  The top-10 drug-target pairs for SARS-COV-2 proteins by ZeroBind

|    | Drug InChl Key | Protein name | Affinity kcal/mole |
|----|----------------|--------------|--------------------|
| 1  | MCQZAGJMVRHLNR-QRWMCTBCSA-N | nucleocapsid protein N-terminal RNA binding domain | -8.2 |
| 2  | VHGZTJJXZVYRMV-UHFFFAOYSA-N | ORF7A encoded accessory protein | -6.9 |
| 3  | IXUYOKHQKZFALL-UHFFFAOYSA-N | ORF7A encoded accessory protein | -7.8 |
| 4  | BBGGFNJGPUEVOH-UHFFFAOYSA-N | u1S2q 1-RBD Up Spike Protein Trimer | -6.9 |
| 5  | OLTVRSUIOUTBRQ-UHFFFAOYSA-N | ORF3a | -5.8 |
| 6  | DETCTMVFPAAUFQ-UHFFFAOYSA-N | main protease | -8.0 |
| 7  | PCVZZJWGMJBSKX-FBMWCMRBSA-N | Orf9b | -8.2 |
| 8  | FDPSDGIMOWDYAV-NDEPHWFRSA-N | u1S2q 1-RBD Up Spike Protein Trimer | -6.5 |
| 9  | VZBSCWDKCMOJCR-UHFFFAOYSA-N | ORF8 accessory protein | -8.4 |
| 10 | FZHRJVXERYVRSK-BZSNNMDCSA-N | Main Protease (Mpro) H172Y Mutant | -7.5 |

-8.4 kcal/mol

-5.8 kcal/mol

**Fig. 4 | ZeroBind predicts binding drugs for SARS-COV-2 proteins. a** Top 10 drug-target binding pairs of SARS-COV-2 proteins. **b** The drug-target binding complex between the drug InChl Key VZBSCWDKCMOJCR-UHFFFAOYSA-N and SARS-CoV-2 ORF8 protein. The green part represents the main part of the protein, the blue part represents the binding drug and the red part represent potential binding sites consisting of the residue name and number. **c** The drug-target binding

complex between the drug InChl Key OLTVRSUIOUTBRQ-UHFFFAOYSA-N and SARS-CoV-2 ORF3a protein. The purple part and blue part represents the two main chains of ORF3a protein, the green part represents the binding drug and the red part represent potential binding sites consisting of the residue name and number. Source data are provided as a Source Data file.

module to find the IB-subgraph, indicating the embedding of the whole protein is less effective than the subgraph embedding. The potential reason is that redundant graph information exists in the whole protein graph, further indicating that the molecule binds to a binding pocket instead of the whole protein. ZeroBind$^{attention-}$ performs slightly worse than ZeroBind, indicating the necessity of task adaptive self-attention module to balance different DTI tasks for different proteins. We also see a performance decline for ZeroBind$^{sampling-}$, due to the lack of the value-generated nonbinding annotations by network-based negative samplings that is able to alleviate the potential short-cut learning. For ZeroBind$^{GIN}$, there is no significant difference in the performance compared with ZeroBind, which indicates that the backbone GNN has a smaller impact on the performance than other modules in ZeroBind. Compared to randomly selected binding pockets, ZeroBind$^{random}$ performs worse than ZeroBind, further validating the effectiveness of the SIB module for detecting potential binding pockets. Moreover, we can see that ZeroBind$^{random}$ with randomly selected pockets performs worse than ZeroBind$^{SIB-}$ without the IB-subgraph module, indicating that initially inaccurate binding pockets guides the model to not locate true binding pockets, resulting in a wrong DTI prediction.

## Discussion

The interaction of proteins with drug molecules is an important research topic, especially in the face of proteins and drug molecules not seen in the training set. Considering the information of proteins and molecules simultaneously is an under-explored idea to solve this problem. In this study, we formulate the drug-target interaction (DTI) prediction as a meta-learning task and propose a meta-learning framework called ZeroBind to solve the generalization problem of unseen proteins and drugs in DTI. Specifically, a meta-learning task is defined as binding drug predictions of a specific protein. The results demonstrate that ZeroBind outperforms existing methods in zero-shot and few-shot scenarios. In addition, the subgraphs learned by SIB align well with binding pockets in the protein, where randomly selected residues as binding pockets almost do not overlap with true binding pockets. Furthermore, we validate ZeroBind's performance in a real-world scenario, where it predicts drug-target bindings for SARS-COV-2.

Due to the rapid development of the research on GNNs, proteins and molecules can be encoded in a more natural form than sequences in previous studies. In addition, the meta-learning strategy also provides a more precise way of delineating the protein-specific DTI task space, which is also consistent with the experimental workflow for proteins in real drug experiments. However, ZeroBind also has some limitations, such as the difficulty of meta-learning training, where the training procedure is complex and prone to instability.

The IB-subgraph method offers an interpretable ability of the model for understanding representation learning. In ZeroBind, the potential binding pockets are automatically detected with the weakly supervised IB-subgraph method that does not use binding pocket annotations as labels. To the best of our knowledge, still no published methods use IB-subgraph or other subgraph-based methods to identify potential binding pockets in the proteins, existing subgraph-based methods are focused on drug molecules. In the current study, we do not incorporate real binding pocket information into model training, since the binding pocket data is far less than the DTI data, and the protein structures predicted by AlphaFold2 still have no known binding pocket annotations. Of the proteins in the benchmark dataset, only 535 proteins have a part of known binding pockets. Thus, it is difficult to train the SIB module in ZeroBind for all proteins to detect binding pockets using the local binding pocket labels. Instead, we train the SIB module of ZeroBind using the global DTI binding labels, which potentially guides the SIB module to locate the potential binding pockets in proteins. As shown in our experiment, ZeroBind with true binding pockets yields better performance. A future update of ZeroBind is expected to take true binding pocket information into the training process to make a more accurate fitting to the DTI problem, if we can collect more true binding pocket data.

Furthermore, the base model in ZeroBind is GCN and there have been more advanced neural architectures for protein-molecule binding, such as SE(3)-equivariant GNN used in EquiBind[43]. In future work, we expect to investigate more advanced GNNs in ZeroBind.

## Methods

### Dataset generation and augmentation

BindingDB[12] is a public database of DTI interactions, which deposits binding affinity data between drugs (drug-like molecules) and target proteins. It currently contains over 2,600,000 experimentally determined binding affinities of protein-drug complexes between over 8000 protein targets and over 1,100,000 small molecules.

To create the training and test datasets for ZeroBind, we apply several filtering and preprocessing steps to create a high-quality benchmark dataset. First, data points are filtered with "single protein" for the "target type" attribute, and kinetic constants $K_i$, $K_d$, $IC_{50}$ and $EC_{50}$ for the "standard type" attribute. In addition, all target proteins should be human or human-like proteins, so they are filtered using

## Table 1 | Performance evaluation of ablation studies on ZeroBind

| Metric | Model | Transductive test | Semi-inductive test | Inductive test |
|---|---|---|---|---|
| AUROC | ZeroBind$^{MAML-}$ | 0.8632 ± 0.0071 | 0.7835 ± 0.0063 | 0.6153 ± 0.0031 |
| | ZeroBind$^{SIB-}$ | 0.8556 ± 0.0056 | 0.8018 ± 0.0121 | 0.7122 ± 0.0042 |
| | ZeroBind$^{attention-}$ | 0.9057 ± 0.0078 | 0.8352 ± 0.0020 | 0.7585 ± 0.0026 |
| | ZeroBind$^{sampling-}$ | 0.9066 ± 0.0018 | 0.8280 ± 0.0031 | 0.7852 ± 0.0043 |
| | ZeroBind$^{GIN}$ | 0.9412 ± 0.0050 | 0.8652 ± 0.0046 | 0.8025 ± 0.0027 |
| | ZeroBind$^{random}$ | 0.8865 ± 0.0017 | 0.8254 ± 0.0082 | 0.7562 ± 0.0031 |
| | **ZeroBind** | **0.9521** ± 0.0038 | **0.8681** ± 0.0065 | **0.8139** ± 0.0045 |
| AUPRC | ZeroBind$^{MAML-}$ | 0.9540 ± 0.0044 | 0.9115 ± 0.0010 | 0.8547 ± 0.0027 |
| | ZeroBind$^{SIB-}$ | 0.9385 ± 0.0117 | 0.9375 ± 0.0028 | 0.9242 ± 0.0058 |
| | ZeroBind$^{attention-}$ | 0.9789 ± 0.0050 | 0.9742 ± 0.0065 | 0.9408 ± 0.0061 |
| | ZeroBind$^{sampling-}$ | 0.9782 ± 0.0027 | 0.9645 ± 0.0063 | 0.9655 ± 0.0057 |
| | ZeroBind$^{GIN}$ | 0.9831 ± 0.0043 | 0.9850 ± 0.0058 | 0.9817 ± 0.0062 |
| | ZeroBind$^{random}$ | 0.9635 ± 0.0056 | 0.9375 ± 0.0151 | 0.9288 ± 0.0071 |
| | **ZeroBind** | **0.9896** ± 0.0013 | **0.9880** ± 0.0062 | **0.9872** ± 0.0020 |

The average is reported after performing each experiment five times, along with the standard deviation. The bold face indicates the method is the best across the compared methods.

"Homo sapiens" for "Target Source Organism" attribute. After excluding proteins that don't have SwissProt name and molecules that cannot be handled by RDKit[44], 1,500,000 protein-drug pairs were collected. We use the threshold in AI-bind[9], which treats kinetic constants $K_i$, $K_d$, $IC_{50}$ and $EC_{50}$ <1000 nM as positive samples and >$10^6$ nM as negative samples.

To demonstrate the effectiveness of ZeroBind, we construct three independent test sets for model evaluation through protein sequence similarity clustering by cd-hit and molecule scaffold split: (1) Transductive test set, where molecules with the same scaffold and proteins with the same cluster are in the training set, but their interactions do not exist in the training set; (2) Semi-inductive test set, where the proteins in the same cluster are in the training set, but the molecules with the same scaffold are not; (3) Inductive test set, where molecules with the same scaffolds and proteins in the same clusters are not in the training set.

We first use cd-hit, a widely used program for clustering protein sequences, to cluster 1603 proteins into 1101 clusters with a similarity threshold 0.4. Through comparing molecule scaffolds, we split the molecules into the training molecule set and test molecule set with a training ratio of 0.95, and ensure that these two sets do not have overlapping scaffolds. As the meta-learning-based framework requires sufficient data, we first divide the clusters that have any protein with the number of associated molecules <20 into the test clusters, and the remaining clusters as the training clusters. Then, we construct the training set using 95% of the proteins in the training clusters and SMILES in the training molecules set. The remaining 5% of the proteins in the training clusters and smiles in the training molecules set are used as the Transductive test set. Finally, the proteins and smiles both in the test clusters and molecule set are further used as the Inductive test set, and the rest data are used as the Semi-inductive test set. During the model training, the cross-validation approach is applied for model optimization.

In addition, we construct another few-shot test set with combining the Semi-inductive test set and the Inductive test set to evaluate the few-shot learning power of ZeroBind. Then, we randomly select 5 positive and 5 negative DTIs of each protein as the few-shot fine-tuning set, and the rest DTIs of each protein as the few-shot test set. Proteins that don't have enough positive or negative DTIs are excluded from the few-shot fine-tuning set and the few-shot test set. The details of the training set and four test sets are given in Table 2.

We observe there exists data imbalance in the training set, and most proteins have a high positive ratio. AI-bind[9] demonstrates that the annotation imbalance causes the network to learn topological shortcuts instead of the binding patterns of drug-target interactions. Similar to AI-Bind[9], we use the network-based negative sampling of training datasets as data augmentation to alleviate the annotation imbalance. Specifically, we construct a bipartite drug-target network. The bipartite network is illustrated in Fig. 1a. Dijkstra's Algorithm is used to find the shortest path distances between any pairs of nodes in the network, and we consider the node pairs with the shortest path distance ≥ 7 in the network as non-binding pairs. After this processing, the annotation imbalance was slightly relieved as shown in Fig. 1b, c.

The 3D structures of 996 proteins in this study are downloaded in PDB format from the RCSB Protein Data Bank (https://www.rcsb.org)[45], and the rest 3D structures of 635 proteins are predicted by AlphaFold2[46]. In addition, we download the true binding pocket information containing pocket residuals and pocket 3D coordinate locations from the PDBbind database, which contains binding pockets for 14,336 DTIs.

## Graph construction for proteins and drugs
In this section, we first introduce the construction of drug graphs and protein graphs, and then give formal definitions of the DTI task and its derivatives, zero-shot DTI predicting task and few-shot DTI predicting task.

**Table 2 | The details of the training set and four test sets**

| Dataset | Number of proteins | Number of drugs | Number of interactions |
|---|---|---|---|
| Training set | 426 | 248,997 | 392,032 |
| Transductive test set | 392 | 16,946 | 17,314 |
| Semi-inductive test set | 1575 | 159,182 | 228,320 |
| Inductive test set | 688 | 7261 | 10,165 |
| Few-shot test set | 1253 | 127,665 | 160,948 |

After using RDkit[44] to construct a molecule from a SMILES[27] string, we apply the encoding format used in Open Graph Benchmark (OGB) datasets[47] to obtain molecular representation. Specifically, atom chemical features consist of atomic number, chirality, atomic formal charge, number of hydrogen atoms attached, radical electrons number, hybridization type, and aromatic. Geometric features consist of atom degree, and a binary value that whether in the ring is used for encoding node representation. Edge features consist of bond type, bond stereo and a binary value that whether bond is conjugated.

Definition 1: We define a drug as a graph denoted as $G_m = \{V_m, E_m\}$, where $V_m$ is the node set representing the atoms of a molecule and $E_m$ is the edge set indicating which pair of nodes are connected.

Definition 2: We define a protein graph as $G_p = \{V_p, E_p\}$ from the protein 3D structure, where $V_p$ is the node set of the residues, $E_p$ is the edge set indicating which pair of residue nodes are connected. In this study, if the Euclidean distance between two residues in 3D structure space is less than 8 angstroms (Å)[48], the two residue nodes are connected with one edge in the protein graph.

For proteins without known 3D structures, we use AlphaFold2[46] to predict their structures as complementary. The node features of the protein graph are initialized with pre-trained ESM-2[49] emebddings, a general-purpose protein language model, of the residues to incorporate prior knowledge. The edge features of the protein graph are the Euclidean distance between the pairs of nodes. The whole process of constructing a protein graph is shown in Fig. 5.

In this work, we train a DTI task per protein.

Definition 3: We define a DTI task $T_p$ from a task set $\mathscr{T}$ for meta learning as $T_p = \{p, M_p\}$, where $p$ is a protein in the protein set $P$ and $M_p$ is the drug set of its corresponding binding molecules and non-binding molecules.

Definition 4: We define the zero-shot DTI predicting task inherited from DTI task, which represents that the meta-model is directly used for prediction without fine-tuning with samples of the protein in meta-learning test set.

Definition 5: We define the few-shot DTI predicting task inherited from DTI task, which uses the protein-specific model for prediction after fine-tuning with samples of the protein in meta-testing.

The few-shot DTI prediction schedule and zero-shot DTI prediction schedule are shown in Fig. 5c.

## Meta-learning setup in ZeroBind
The meta-learning framework was originally proposed to learn general knowledge across multiple related tasks within a distribution and used this general experience to quickly adapt to additional tasks and improve prediction performance[50]. There are two main types of meta-learning framework: (1) Gradient-based methods: The classic gradient-based method MAML[51] uses a meta-learner to learn a good initialization by summing up multiple task losses and updating the parameters across tasks. Therefore, MAML could achieve both high accuracy and speed for generalization; (2) Metric-based methods: The prototypical network[52] is a classic metric-based meta-learning algorithm that directly trains the vector representation (i.e., prototypes) of each category. Once a good feature extractor is trained, the category of a new sample is determined by its closest prototype in the vector space.

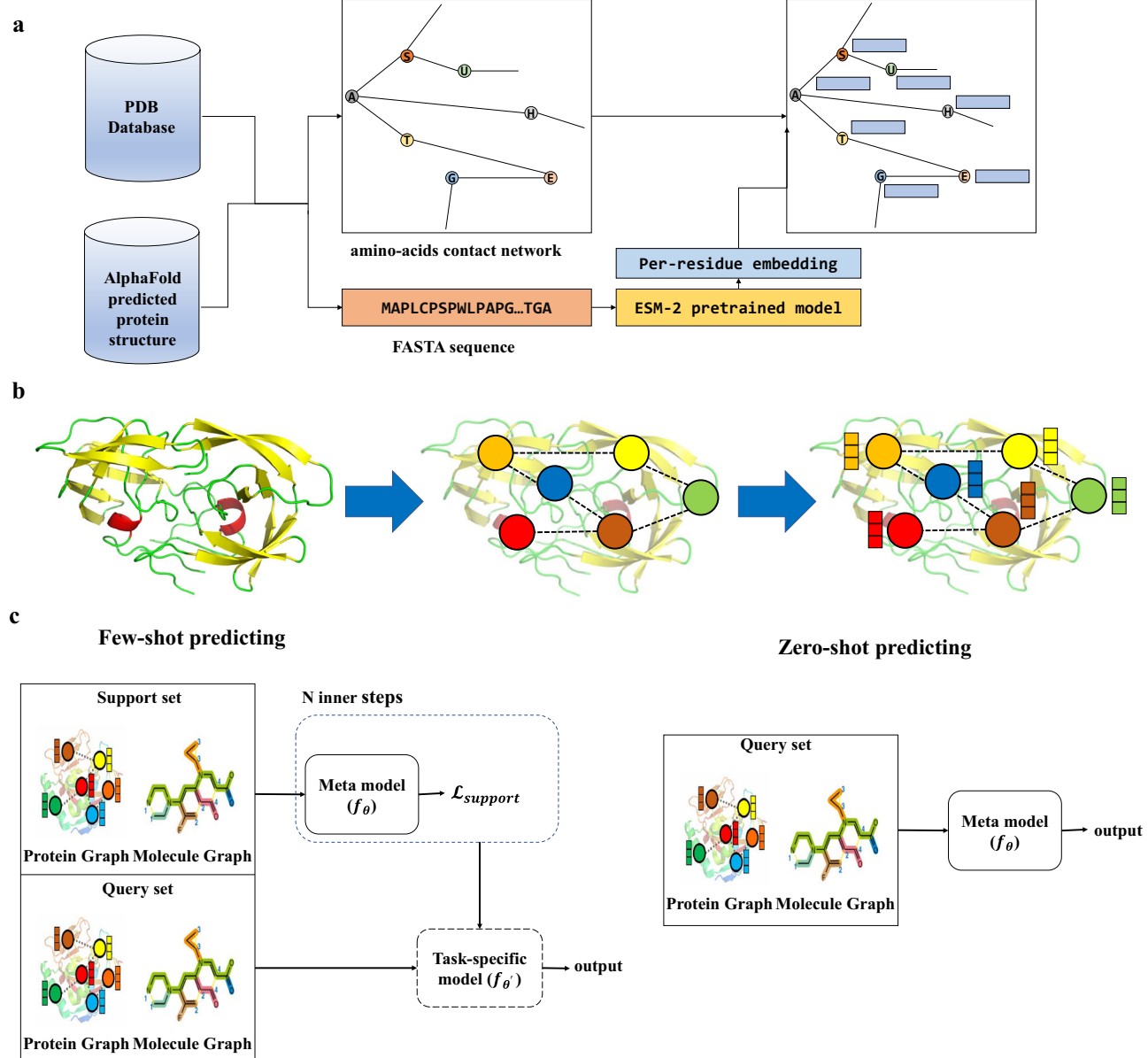

**Fig. 5 | The data processing of ZeroBind. a** The process of constructing a protein graph from the protein 3D structure. Instead of using peptide bonds as edges, we connect two residues with the cutoff distance < 8 angstroms (Å), where edges and node features are extracted using ESM-2 pretrained model. **b** An example of the protein graph construction from the protein 3D structure. **c** The few-shot DTI prediction process and zero-shot DTI prediction process.

In our study, we apply gradient-based methods MAML as the training strategy. Besides, we follow MAML++[37] to make some improvements to stabilize the training process of MAML.

In the details of ZeroBind, several techniques including multi-step loss optimization (MSL), learning per-layer per-step learning rates and gradient directions (LSLR), subgraph information bottleneck (SIB), and task adaptive self-attention are applied to improve the performance of ZeroBind. We build the zero-shot learning framework based on MAML. First, we randomly initialize the network parameter $\theta$. Given training tasks (proteins) sampled from the task distribution $\mathcal{T}$, MAML aims to learn good initial parameters that can quickly adapt to additional tasks. For a pair of a protein and drug, we sample a 2-way, k-shot, m-query training task. Of these three hyperparameters, 2-way is set up since there are only two types of labels of binding and nonbinding in the classification task. In the original MAML framework, the model first trains $k$ data samples of each label for several inner steps to learn a task-specific model and then tests on $m$ data samples. After obtaining the loss for each task, MAML applies the gradient descent with the sum or average of the loss across all tasks. Here we refer the training task and test task as the support set and query set, respectively. However, MAML suffers from instability during training and is sensitive to neural network architectures and a large number of hyperparameters. Therefore, we follow MAML++[37] training procedure, which leverages multi-step loss optimization (MSL) and learning per-layer per-step learning rates and gradient directions (LSLR) in MAML.

Specifically, the multi-step loss optimization calculates the loss of the query set after each inner step, and optimizes the base network with the support set after all inner step optimization is completed. More formally, we optimize the model parameters as follows:

$$\theta = \theta - \alpha \nabla_\theta \sum_{b=1}^{B} \sum_{i=0}^{N} v_i \mathcal{L}_{T_b}\left(f_{\theta_i^b}\right) \quad (1)$$

where $\alpha$ is the learning rate, $\mathcal{L}_{T_b}\left(f_{\theta_i^b}\right)$ denotes the loss of the query set after every inner step optimization with the support step and $v_i$ denotes the importance of step $i$ generated by the number of inner steps.

## BOX 1

## ZeroBind training process

Require: $\left\{\mathbf{G_m^{\tau}}, \mathbf{G_p^{\tau}}, \mathbf{Y^{\tau}}\right\}$ : training data;
While not done do:
 Sample batch of tasks $\mathbf{T}_{\tau} \sim \mathbf{p}(\tau)$
 For all $\mathbf{T}_{\tau}$ do
 Sample k examples as support sets of $\mathbf{T}_{\tau}$ $\left\{\mathbf{G_{mn}^{\tau}}, \mathbf{G_{pn}^{\tau}}, \mathbf{Y_n^{\tau}}\right\}_{\mathbf{n=1}}^{\mathbf{k}} \in \left(\mathbf{G_m^{\tau}}, \mathbf{G_p^{\tau}}, \mathbf{Y^{\tau}}\right)$
 Sample m examples as query sets of $\mathbf{T}_{\tau}$ $\left\{\mathbf{G_{mn}^{\tau}}, \mathbf{G_{pn}^{\tau}}, \mathbf{Y_n^{\tau}}\right\}_{\mathbf{n=1}}^{\mathbf{m}} \in \left(\mathbf{G_m^{\tau}}, \mathbf{G_p^{\tau}}, \mathbf{Y^{\tau}}\right)$
 $\theta_{\mathbf{base}} \leftarrow \theta$
 For i=1 to inner steps do:
 Calculate the inner step loss weight $\omega_{\mathbf{i}}$
 $\left\{\mathbf{y_{in}^{\tau}}\right\}_{\mathbf{n=1}}^{\mathbf{k}} = \mathbf{BaseModel}\left(\left\{\mathbf{G_{Pn}^{\tau}}\right\}_{\mathbf{n=1}}^{\mathbf{k}}, \left\{\mathbf{G_{Mn}^{\tau}}\right\}_{\mathbf{n=1}}^{\mathbf{k}}; \theta_{\mathbf{base}}\right)$
 $\mathscr{L}_{\mathbf{i}}^{\tau} \leftarrow \mathbf{Eq}.(11) \mathbf{with} \left\{\mathbf{y_{in}^{\tau}}\right\}_{\mathbf{n=1}}^{\mathbf{k}}$
 $\theta_{\mathbf{base}}$ optimize with $\left\{\mathbf{y_{in}^{\tau}}\right\}_{\mathbf{n=1}}^{\mathbf{k}}$
 $\left\{\mathbf{y_{in}^{\tau}}\right\}_{\mathbf{n=1}}^{\mathbf{m}} = \mathbf{BaseModel}\left(\left\{\mathbf{G_{Pn}^{\tau}}\right\}_{\mathbf{n=1}}^{\mathbf{m}}, \left\{\mathbf{G_{Mn}^{\tau}}\right\}_{\mathbf{n=1}}^{\mathbf{m}}; \theta_{\mathbf{base}}\right)$
 $\mathscr{L}_{\mathbf{i}}^{\tau} \leftarrow \mathbf{Eq}.(11) \mathbf{with} \left\{\mathbf{y_{in}^{\tau}}\right\}_{\mathbf{n=1}}^{\mathbf{m}}$
 End for
 $\mathscr{L}^{\tau} = \left\{\omega_{\mathbf{i}}\right\}_{\mathbf{i=1}}^{\mathbf{N}} \cdot \left\{\mathscr{L}_{\mathbf{i}}^{\tau}\right\}_{\mathbf{i=1}}^{\mathbf{N}}$
 End for
 $\left\{\eta_{\mathbf{i}}\right\}_{\mathbf{i=1}}^{\mathbf{N}} \leftarrow \mathbf{Eq}.(14)$
 $\theta \leftarrow \theta - \alpha \nabla_{\theta} \sum_{\mathbf{T}_{\tau} \sim \mathbf{p}(\tau)} \eta_{\mathbf{i}} \cdot \mathscr{L}^{\tau}$
End while

## BOX 2

## BaseModel forward process

Require: $\left\{\mathbf{G_P}, \mathbf{G_M}\right\}$ : training data;
$\mathbf{G_P} = \mathbf{GNN_P}\left(\mathbf{G_P}; \theta_P\right)$
$\mathbf{G_M} = \mathbf{GNN_M}\left(\mathbf{G_M}; \theta_M\right)$
$\mathbf{Z} \leftarrow \mathbf{Eq}.(4) \mathbf{with}\left(\mathbf{G_P}, \mathbf{G_M}\right)$
$\mathbf{G_{Psub}} \leftarrow \mathbf{Eq}.(5) \mathbf{with}\left(\mathbf{Z}, \mathbf{G_P}\right)$
For i=1 to num steps do
 $\mathscr{L}_{\mathbf{MI-pro}} \leftarrow \mathbf{Eq}.(8) \mathbf{with}\left(\mathbf{G_P}, \mathbf{G_{Psub}}; \varphi_2\right)$
 $\mathscr{L}_{\mathbf{MSE}} \leftarrow \mathbf{Eq}.(6) \mathbf{with}\ \mathbf{Z}$
 $\mathscr{L}_{\mathbf{cls}} \leftarrow \mathbf{Eq}.(10) \mathbf{with}\left(\mathbf{G_m}, \mathbf{G_{Psub}}; \mathbf{Y}\right)$
 $\mathscr{L}_{\mathbf{Base}} \leftarrow \mathbf{Eq}.(11) \mathbf{with}\left(\mathscr{L}_{\mathbf{cls}}, \mathscr{L}_{\mathbf{MSE}}, \mathscr{L}_{\mathbf{MI-pro}}\right)$

The LSLR module sets the learning rate to be learnable parameters for each layer at each inner step. Without adding much computation, different learning rates are automatically learned for each layer at each step, which may help alleviate overfitting.

To summarize the meta-learning procedure in ZeroBind (Boxes 1 and 2), the model is first updated to task-specific models using the support set of each task, and then calculates the loss of the query set of the task. After repeating N inner steps, the losses of all query sets are weighted average by $\{\eta_{\mathbf{T}_i}\}_{\mathbf{i=1}}^{\mathbf{N}}$, and are used to optimize the meta-model by gradient descent. After training with a sufficient number of samples, the learned model has a good capability of predicting DTI for unseen molecules and proteins or quickly adapting to additional protein tasks with a few training samples. The meta-learning procedure of ZeroBind is shown in Fig. 1d.

### Architecture of base model in ZeroBind

Figure 1e shows the architecture of the base model in ZeroBind, which mainly consists of three modules: a GNN module to obtain the embedding of molecules and proteins, a SIB module to find the most predictive subgraph as the binding pocket in the protein, and a dense module with concatenating protein subgraph representation and molecular representation to score the interactions.

Graph neural network-based representation learning for proteins and drugs. After constructing drug and protein graphs for a given protein-drug pair, we first embed the molecule atom features to vector space with randomly initialized parameters. We denote the node embedding of the molecule as $\mathbf{X_m}\mathbb{R}^{N_m * D_m}$, and also as the initial node embedding of GNN, denoted as $\mathbf{G_m^{(0)}}$, where $N_m$ is the number of nodes and $D_m$ is the dimension of the node embeddings. Here, we denote the graph convolutional network (GCN) based molecule backbone as $\mathbf{GCN_m}$, which is used to learn the graph embedding from the molecule graph. Then, the $l$-th layer output of $\mathbf{GCN_m}$ can be formulated as:

$$G_m^{(l)} = RELU\left(\hat{A}_m G_m^{(l-1)} W_m^{(l-1)} + b^{(l-1)}\right) \tag{2}$$

where $\hat{\mathbf{A}}_{\mathbf{m}}$ represents the adjacent matrix of a graph, $\mathbf{G_m^{(l-1)}}$ denotes the node embedding of $(l-1)$-th layer and $\mathbf{W_m^{(l-1)}}$ is the learnable weight matrix.

For protein graphs, we initialize the node embedding $\mathbf{G_p^{(0)}}$ of a protein graph with ESM-2, a pre-trained general-purpose protein embedding. Similarly to the molecule graph, a GCN-based backbone $\mathbf{GCN_p}$ is used to learn the graph embedding from the protein graphs. Then, the $l$-th layer output of $\mathbf{GCN_p}$ can be formulated as:

$$G_p^{(l)} = RELU\left(\hat{A}_p G_p^{(l-1)} W_p^{(l-1)} + b^{(l-1)}\right) \tag{3}$$

Subgraph learning in ZeroBind for potential binding pockets in proteins. Considering that a molecule binds to a binding pocket in the protein instead of the whole protein, after the protein backbone GCN, we apply a model-agnostic SIB module[53] to identify the most interpretable subgraph with the most crucial information associated with the DTI task, where the learned subgraph corresponds to the binding pocket on the protein graph. The SIB module is proposed to recognize a compressed subgraph named IB-subgraph under the information bottleneck (IB) principle. The IB-subgraph could also eliminate the noisy and redundant graph information. In our DTI task, the DTI score

is largely determined by the protein pocket features, which are the subgraph information bottleneck of the protein graph or the potential drug binding sites in the protein.

The SIB module contains a subgraph generator for node assignment and a dense layer to guarantee that the generated subgraph matches the IB-subgraph. The subgraph generator is a Multi-layer Perceptron (MLP), which takes the protein node embedding and the molecule embedding as the input and outputs the node assignment matrix $\mathbf{Z}$, which is formulated as follows:

$$Z = Softmax\left(MLP\left(concatenate\left(G_p^{(l)}, G_m^{(l)}\right); \varphi_1\right)\right) \tag{4}$$

$$G_{sub} = Z^T G_p^{(l)}[0] \tag{5}$$

where the output of the MLP is a $n \times 2$ matrix with the number of nodes n. After applying *Softmax* on the output of MLP layer, each row of $\mathbf{Z}$ is the probability of the corresponding node belonging to the $\mathbf{G_{sub}}$ and $\mathbf{\bar{G}_{sub}}$, denoted as $[p(h_i \in G_{sub}|h_i), p(h_i \in \bar{G}_{sub}|h_i)]$, where $\mathbf{G_{sub}}$ represents the generated subgraph of protein graph $\mathbf{G_P}$ and $\mathbf{\bar{G}_{sub}}$ represents the leftover subgraph of protein graph. After training with a sufficient number of training samples, the learned node assignment matrix $\mathbf{Z}$ is supposed to approach to 0 or 1. Thus, the generated IB-subgraph embedding can be obtained by taking the first row of the product between $\mathbf{Z^T}$ and node embedding.

To stabilize the training process of the subgraph generation and separate $p(h_i \in G_{sub}|h_i)$ from $p(h_i \in \bar{G}_{sub}|h_i)$, we apply a connectivity loss as follows:

$$\mathcal{L}_{MSE} = MSE\left(diag\left(Norm\left(Z^T A Z\right)\right) - diag(I_2)\right) \tag{6}$$

where $Norm(\cdot)$ is the row normalization, $\mathbf{A}$ is the adjacency matrix of protein graph $\mathbf{G_P}$ and $\mathbf{I_2}$ is a $2 \times 2$ identity matrix. Minimizing $\mathcal{L}_{MSE}$ could encourages $[p(h_i \in G_{sub}|h_i), p(h_i \in \bar{G}_{sub}|h_i)] \rightarrow [0,1]/[1,0]$, which results in a distinctive $\mathbf{Z}$ to stabilize the training process and a compact topology in the subgraph.

Formally, the subgraph information bottleneck seeks the IB-subgraph by optimizing the following objective function:

$$\max_{G_{sub}} I(Y, G_{sub}) - \beta I(G, G_{sub}) \tag{7}$$

where $\beta$ is a weight, $Y$ represents the label associated with $\mathbf{G}$, and $\mathbf{G}$ represents the original graph. $I(Y, G_{sub})$ and $I(G, G_{sub})$ respectively represents the mutual information between $(Y, G_{sub})$ and $(G, G_{sub})$, and $\mathbf{G_{sub}}$ represents the IB-subgraph embedding.

To optimize the objective function, the lower bound of the first term is formulated as the opposite number of classification loss between the ground truth label and predicted label with subgraph embedding. By minimizing the classification loss, the lower bound of $I(Y, G_{sub})$ achieves the maximum value. For the second term, the DONSKER-VARADHAN representation[54] of the KL-divergence is applied to approximate the upper bound of $I(G, G_{sub})$. The approximation of $I(G, G_{sub})$ can be formulate as:

$$\max_{\varphi_2} \mathcal{L}_{MI-pro}(\varphi_2, G_{sub}) = \frac{1}{N} \sum_{i=1}^{N} Dense\left(G_i, G_{sub_i}; \varphi_2\right) \\ - \log \frac{1}{N} \sum_{i=1, j \neq i}^{N} e^{Dense\left(G_i, G_{sub_j}; \varphi_2\right)} \tag{8}$$

where $Dense(\bullet)$ is the dense network of several MLP layers with the concatenation of the embedding of $\mathbf{G}$ and $\mathbf{G_{sub}}$ as the input.

To minimize $I(G, G_{sub})$, several inner steps of the maximum optimization of $\mathcal{L}_{MI-pro}$ are applied to minimize the upper bound of $I(G, G_{sub})$.

After obtaining the embedding of the generated protein IB-subgraph, we concatenate the molecular embedding and protein IB-subgraph embedding as the input to the MLP classification layer. The DTI classification loss could be formulated as:

$$\hat{y} = MLP\left(concatenate(G_m, G_{sub}); \theta_{cls}\right) \tag{9}$$

$$\mathcal{L}_{cls} = -y \log \hat{y} - (1-y) \log(1-\hat{y}) \tag{10}$$

The overall loss can be formulated as:

$$\min_{\theta_M, \theta_P, \varphi_1, \varphi_2, \theta_{cls}} \mathcal{L}_{base} = \mathcal{L}_{cls} + \lambda_1 \mathcal{L}_{MSE} + \lambda_2 \mathcal{L}_{MI-pro} \tag{11}$$

$$s.t. \varphi_2^* = \arg \max_{\varphi_2} \mathcal{L}_{MI-pro}$$

where $\lambda_1$ and $\lambda_2$ are weights to balance the importance of different losses, $\theta_M, \theta_P$ are the parameters of $GCN_m$ and $GCN_p$, $\varphi_1$ is the parameters of IB-subgraph generator, $\varphi_2$ is the parameters of the dense network, and $\theta_{cls}$ is the parameters of the MLP layer.

Here, we summarize the training process, the $\mathcal{L}_{cls}$ measures the discrepancy between the estimated distribution and the original data distribution by utilizing the cross-entropy between the predicted values and the ground truth (the DTI labels instead of binding pocket labels). The $\mathcal{L}_{MSE}$, as indicated by Eq. (6), encourages the node assignment matrix $Z$ to approach close to either 1 or 0. The $\mathcal{L}_{MI-pro}$, as indicated by Eq. (8), represents an approximation of $I(G, G_{sub})$ introduced in Eq. (7). To maximize Eq. (7) and achieve subgraph information bottleneck, the lower bound of $I(Y, G_{sub})$ needs to increase as much as possible and the upper bound of $I(G, G_{sub})$ needs to decrease. Therefore, we begin by performing gradient descent on the negative of $\mathcal{L}_{MI-pro}$ within the dense network. The goal is to maximize $\mathcal{L}_{MI-pro}$ and achieve the upper bound of $I(G, G_{sub})$, thus satisfying the constraints of Eq. (11). After that, we utilize gradient descent to optimize Eq. (11). By optimizing $\mathcal{L}_{cls}$, we increase the lower bound of $I(Y, G_{sub})$, and decrease the upper bound of $I(G, G_{sub})$ by optimizing $\mathcal{L}_{MI-pro}$. This approach aims to maximize Eq. (7) and seek the subgraph information bottleneck for detecting potential binding pockets in proteins.

In our task, we aim to find the protein binding pocket under the weakly supervised training process, since the amount of protein binding pockets data is much smaller than the amount of binding data. So we design the IB-subgraph to find the protein binding pockets optimized on the DTI binding labels instead of binding pocket labels in proteins. Considering the protein pockets are determined by both proteins and molecules, we take the concatenation of molecule embedding and protein embedding as input when we generate the node assignment matrix $Z$ in Eq. (4).

**Task adaptive self-attention**

In a traditional meta-learning schedule, *batch_size* tasks are sampled and treated with the same weight when applying mini-batch gradient descent. However, the same weight of multiple tasks can't reflect the importance of different tasks contributing to the meta-model optimization. We further design a task adaptive self-attention module (Fig. 1f) to learn the importance of different tasks automatically. Since each task is a DTI prediction of a specific protein, we take the concatenation of protein subgraph embedding and the average of the embeddings of all molecules in the query set as the task embedding. The self-attention module is:

$$h_{T_b} = concatenate\left(G_p, Mean\left(\left\{G_m^i\right\}_{i=1}^m\right)\right) \tag{12}$$

$$Q = W_Q \left\{ h_{T_b} \right\}_{b=1}^{B}$$
$$K = W_K \left\{ h_{T_b} \right\}_{b=1}^{B} \quad (13)$$
$$V = W_V \left\{ h_{T_b} \right\}_{b=1}^{B}$$

$$\left\{ \eta_{T_b} \right\}_{b=1}^{B} = Attention(Q,K,V) = softmax\left( \frac{QK^T}{\sqrt{d_k}} \right) V \quad (14)$$

$$\mathscr{L}_{all} = \sum_{b=1}^{B} \eta_{T_b} \mathscr{L}_{query,b} \quad (16)$$

where $T_b$ is one of the DTI tasks, $\mathbf{h_{T_b}}$ represents the task embedding, $\mathbf{W_Q}$, $\mathbf{W_K}$ and $\mathbf{W_V}$ are learnable parameter matrices, $\mathbf{Q,K,V}$ are the generated attention matrices, $d_k$ is the dimension of task embedding to normalize the variance of $\mathbf{QK^T}$ and stabilize gradient values during training, and $\boldsymbol{\eta_{T_b}}$ is the generated weight to balance the importance of different task.

MetaDTA[36] uses multi-head cross-attention network to capture the relationships between the support and the query ligands instead of different proteins, where it does not use any protein information. In contrast, ZeroBind utilizes a task adaptive self-attention module to learn the importance for different tasks of proteins, and it focuses on the shared binding patterns among different proteins.

### Experimental settings
We take graph convolutional networks as base graph neural network. In our experiment, the GCN layer number of $\mathbf{GCN_P}$ is set as 4 and we set the embedding dimensions as 1280,512,256,256,256, and the GCN layer number of $\mathbf{GCN_M}$ is set as 3, the embedding dimensions we set are 256,256,256. We set the update step in inner loops as 5. 20 inner steps of the maximum optimization of $\mathscr{L}_{MI-pro}$ are applied. We set the $\lambda_1$ and $\lambda_2$ as 0.05. The learning rate of optimization of $\mathscr{L}_{MI-pro}$ is set at 0.01. The meta learning rate is using simulated annealing algorithm. We use Pytorch to implement the model and run it on a GPU.

### Baseline methods
We compare ZeroBind with multiple baselines to evaluate its advantages, and calculate the area under the receiver operating characteristic (AUROC) and the area under the precision-recall curve (AUPRC) on three independent test sets and one few-shot test set.

1. DeepConv-DTI[11]: DeepConv-DTI trains a CNN to learn the embedding of the neighboring residues and an MLP to learn the molecular fingerprint, which is concatenated to make DTI prediction.
2. GraphDTA[10]: GraphDTA leverages multiple types of GNN models to embed molecular embeddings and a CNN model to embed protein residue features.
3. Deeppurpose[8]: Deeppurpose is a state-of-art deep learning library for DTI prediction with multiple backbone networks. Here we use CNN as the backbone network for learning embeddings of both drugs and proteins.
4. AI-bind[9]: AI-bind uses pre-trained mol2vec and protvec models to initialize the molecule embedding and protein embedding, and makes the DTI predictions by concatenating the molecule embedding and protein embedding.
5. DrugBAN[55]: DrugBAN uses a deep bilinear attention network (BAN) framework with domain adaptation to explicitly learn local pairwise interactions between drugs and targets.

### Reporting summary
Further information on research design is available in the Nature Portfolio Reporting Summary linked to this article.

## Data availability
The online webserver is freely available at http://www.csbio.sjtu.edu.cn/bioinf/ZeroBind/. The benchmark dataset is collected from the original database BindingDB (https://www.bindingdb.org/bind/downloads/BindingDB_All_2D_202311_sdf.zip) and it is freely available at http://www.csbio.sjtu.edu.cn/bioinf/ZeroBind/datasets.html along with the SARS-COV-2 test dataset, and the experimental protein structure data used in this study are downloaded from the RCSB PDB database (https://www.rcsb.org/downloads/) and the predicted structures by AlphaFold are downloaded from AlphaFold Protein Structure Database (https://www.alphafold.ebi.ac.uk/). All PDB and AlphaFold codes can be found on GitHub (https://github.com/myprecioushh/ZeroBind). Source data are provided with this paper.

## Code availability
The source codes of ZeroBind are available on GitHub (https://github.com/myprecioushh/ZeroBind), together with a usage documentation and setup environment.

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

## Acknowledgements

This work was supported by the National Key Research and Development Program of China (No. 2020AAA0107600 to X.P.), the National Natural Science Foundation of China (No. 61725302 and 62073219 to H.B.S., 61903248 to X.P.), and the Science and Technology Commission of Shanghai Municipality (20S11902100 to X.P., 22511104100 to H.B.S.).

## Author contributions

Y.W. contributed to writing the manuscript, data curation, and preparation, implemented the prediction model and performing the experiments, running docking simulation, and writing the manuscript. Y.X. contributed to data preparation and writing the manuscript. J.Y., Y.Y., and H.B.S. provided the guidance on designing the model and experiments and writing the manuscript. X.P. conceived the project, designed the experiments, contributed to data analysis and writing the manuscript.

## Competing interests

The authors declare no competing interests.
