## [Peer Review File · Nature Communications]

REVIEWER COMMENTS

Reviewer #1 (Remarks to the Author):

The paper addresses a significant problem in drug discovery and proposed meta-learning-based subgraph matching method. Each target protein has its own model, adapted from the meta-model to explicitly capture the information from the binding target. The paper is generally easy to follow and well-presented. The authors also conducted intensive experiments to validate the advantage of the proposed method over a couple of state-of-the-art baseline methods.

However, there are a couple of concerns of the paper, listed as follows.

The experimental setup may be problematic. The paper uses random split to separate the whole dataset into training/validation/test sets, which may bring some biases for drug-target interaction problems. The scaffolding split (split training/validation/test set based on scaffolds of the drug and target protein) would be a better split strategy for the drug-target interaction problem, because scaffolding split would make sure that for the data points (drug-target pair) in the test set, both drug and target protein does not appear in training and validation set. Please refers to https://tdcommons.ai/functions/data_split/#scaffold-split for more details.

For empirical studies, it could be great if the authors can show the results of multiple independent runs with different data split and random seeds, which would make results more convincing.

The authors should emphasize the methodology novelty of the paper. What is the difference between the proposed method and the reference [48]? I understand the subgraph information bottleneck is the major methodology novelty.

In your GCN formulation in Equation (2), do you use bias parameters? Based on my experience, removing the bias parameter would degrade the GCN's performance.

For protein embedding, protein can be either represented as a 1D amino acid sequence or a 3D geometric structure, however, GCN can embed 2-dimensional graph only. So, it needs more explanation on how to use GCN to represent the target protein structure.

The whole objective function is a min-max optimization problem, how do you solve this problem? Please elaborate on the details.

I also suggest authors use bold capitalized letters for the matrix and bold lowercase letters for the vector to discriminate it from the scalar.

Reviewer #2 (Remarks to the Author):

Please kindly review the attached file. Thank you!

Key results

The paper presents a novel meta-learning framework for predicting protein-drug interactions, addressing the challenge of learning interactions involving new drugs or proteins. The core architecture of the proposed model is a graph neural network that represents both the drug and protein structures as graphs, incorporating atom features and pre-trained residue embeddings as node features. To capture the binding pocket of the protein, a subgraph information bottleneck technique is employed.

Validity

1. Figure 2b does not provide conclusive evidence regarding the correlation between the number of training proteins and the performance of the models (DeepConv, GraphDTA, DeepPurpose, Albind, and Zerobind). The plot suggests that there may not be a clear relationship between these factors. However, further analysis and statistical tests are needed to draw definitive conclusions.
2. The authors introduce the Jaccard similarity between the learned binding pocket and the true binding pocket. From Figures 2a and 2b, it appears that the false positive rate is relatively high. Changing the binding pocket can indeed alter the binding configuration, leading to different binding affinity or interaction. There is a concern that the Subgraph Information Bottleneck (SIB) module may learn factors other than the binding pocket since the entire process is unsupervised. To address this concern, it would be valuable to investigate the performance when the true binding pocket is provided or when the SIB module is trained in a supervised manner. Additionally, conducting an ablation study could further support the argument for the validity of the SIB module.

Significance

The paper highlights three primary contributions: meta-learning, task adaptive self-attention, and IB-subgraph. However, it should be noted that these methods have been previously employed in the drug-protein prediction task, so the level of novelty may not be particularly high. For instance, MetaDTA [1] has utilized similar techniques for learning and task adaptive self-attention. Additionally, the idea of learning the binding pocket unsupervised through IB-subgraph has been extensively explored, as seen in studies like GEFA [2]. To demonstrate the effectiveness of the proposed

approach, the authors should compare their results with these baseline methods and clearly showcase the improvements achieved through their design.

Data and methodology

It is important for the authors to include details about the data used in the COVID drug section. Without this information, it becomes difficult to verify the results of the experiment conducted in that particular domain.

References

[1] Lee, E., Yoo, J., Lee, H., & Hong, S. (2022, April). MetaDTA: Meta-learning-based drug-target binding affinity prediction. In ICLR2022 Machine Learning for Drug Discovery.

[2] Nguyen, T. M., Nguyen, T., Le, T. M., & Tran, T. (2021). GEFA: early fusion approach in drug-target affinity prediction. *IEEE/ACM Transactions on Computational Biology and Bioinformatics*, 19(2), 718-728.

Your expertise

I have a high level of confidence in my review of the drug-protein modelling, data, and experiment, and I would rate it 4 out of 5. However, I am less confident in the usage of meta-learning in the drug-protein context, and I would give it a score of 2 out of 5.

We carefully addressed all the comments. The changes have been highlighted in red in the revised manuscript. The detailed responses to the reviewers are attached below and highlighted in blue. We highly appreciate the constructive comments, which are very helpful in strengthening this paper.

Response to reviewer 1

The paper addresses a significant problem in drug discovery and proposed meta-learning-based subgraph matching method. Each target protein has its own model, adapted from the meta-model to explicitly capture the information from the binding target. The paper is generally easy to follow and well-presented. The authors also conducted intensive experiments to validate the advantage of the proposed method over a couple of state-of-the-art baseline methods.

A: Thanks for your comments on our manuscript. Below comments are constructive to strengthen the manuscript. We have revised the manuscript according to the below comments.

1. Q: The experimental setup may be problematic. The paper uses random split to separate the whole dataset into training/validation/test sets, which may bring some biases for drug-target interaction problems. The scaffolding split (split training/validation/test set based on scaffolds of the drug and target protein) would be a better split strategy for the drug-target interaction problem, because scaffolding split would make sure that for the data points (drug-target pair) in the test set, both drug and target protein does not appear in training and validation set. Please refers to https://tdcommons.ai/functions/data_split/#scaffold-split for more details.

A: Thanks for your comments. We have updated our experiments with scaffold split setup. The updated results are shown in Results part and the scaffold split setting is presented in Dataset Generation and Augmentation section of the revised manuscript.

We construct three independent test sets for model evaluation through protein sequence similarity clustering by cd-hit and molecule scaffold split: 1) Transductive test set, where molecules with the same scaffold and proteins with the same cluster are in the training set, but their interactions do not exist in the training set; 2) Semi-inductive test set, where the proteins in the same cluster are in the training set, but the molecules with the same scaffold are not; 3) Inductive test set, where molecules with the same scaffolds and proteins in the same clusters are not in the training set.

2. Q: For empirical studies, it could be great if the authors can show the results of multiple independent runs with different data split and random seeds, which would make results more convincing.

A: We have conducted five independent experiments using different random seeds. The updated average results are shown in Table S1 and Figure 2a of the revised manuscript, and the t-test p-values are also significant for performance comparison.

3. Q: The authors should emphasize the methodology novelty of the paper. What is the difference between the proposed method and the reference [48]? I understand the subgraph information bottleneck is the major methodology novelty.

A: Thanks for your comments. We introduce our further development of reference [48] in our

DTI prediction. Reference [48] aims to find the subgraph information bottleneck, i.e. the maximally informative and compressive subgraphs regarding the graph label. In our task, we aim to find the protein binding pockets under the unsupervised training process that does not use binding pocket labels during model training, since the amount of protein pockets data is much smaller than the amount of DTI binding data. So we design the IB-subgraph to find the protein pocket by optimizing IB-Subgraph on the binding DTI labels instead of the binding pocket labels. Considering the protein pockets are determined by both proteins and molecules, we take the concatenation of molecule embedding and protein embedding as input when we generate the node assignment matrix Z in Equation (4). In addition, the $I(Y, G_{sub})$ in Equation (8) is formulated as the opposite value of \mathcal{L}_{cls} which corresponds to the cross-entropy loss between the predicted binding DTI label and the ground truth. It is important to note that our \mathcal{L}_{cls} does not refer to the cross-entropy loss between the predicted binding pocket labels and the true binding pocket labels, as described in reference [48]. We claim these points in section “Subgraph learning in ZeroBind for potential binding pockets in proteins” of the revised manuscript.

4. Q: In your GCN formulation in Equation (2), do you use bias parameters? Based on my experience, removing the bias parameter would degrade the GCN’s performance.

A: We have used the bias parameters in Equation (2) and other similar equations. We have revised the formulations in the revised manuscript.

5. Q: For protein embedding, protein can be either represented as a 1D amino acid sequence or a 3D geometric structure, however, GCN can embed 2-dimensional graph only. So, it needs more explanation on how to use GCN to represent the target protein structure.

A: Thanks for your comments. As you can see in “Graph construction for proteins and drugs” section and Figure 5(b), we encode protein 3D structure information into a graph by taking residues as graph nodes and whether the distance between residues in 3D structure space exceeds a threshold as graph edges. If the distance between two residues is less than 8 angstroms (\AA)¹, the two residue nodes are connected with one edge. For node features of the protein graph, we use the pre-trained embeddings of residues as the node features. We updated the details in Definition 2 of section “Graph construction for proteins and drugs” in the revised manuscript.

6. Q: The whole objective function is a min-max optimization problem, how do you solve this problem? Please elaborate on the details.

A: To accurately estimate the predicted values, Equation (11) serves as the overall loss function, which is optimized using the gradient descent during the model training:

$$\min_{\theta_M, \theta_P, \varphi_1, \varphi_2, \theta_{cls}} \mathcal{L}_{base} = \mathcal{L}_{cls} + \lambda_1 \mathcal{L}_{MSE} + \lambda_2 \mathcal{L}_{MI-pro} \quad (11)$$

$$s. t. \varphi_2^* = \arg \max_{\varphi_2} \mathcal{L}_{MI-pro}$$

The \mathcal{L}_{cls} measures the discrepancy between the estimated distribution and the original data distribution by utilizing the cross-entropy between the predicted values and the ground truth (the DTI labels instead of local binding pocket labels). The \mathcal{L}_{MSE} , as indicated by Equation (6), encourages the node assignment matrix Z to approach either 1 or 0. The \mathcal{L}_{MI-pro} , as

indicated by Equation (8), represents the approximation of the mutual information of G and G_{sub} introduced in Equation 7. The Equation (7) seeks the subgraph information bottleneck by maximum the mutual information of Y and G_{sub} minus the mutual information of G and G_{sub} .

$$\max_{G_{sub}} I(Y, G_{sub}) - \beta I(G, G_{sub}) \quad (7)$$

The lower bound of $I(Y, G_{sub})$ is formulated as the opposite value of \mathcal{L}_{cls} . Therefore, the minimization of \mathcal{L}_{cls} also increases the lower bound of $I(Y, G_{sub})$. The upper bound of $I(G, G_{sub})$ could be approximate by using the DONSKER-VARADHAN representation of the KL-divergence.

$$\max_{\varphi_2} \mathcal{L}_{MI-pro}(\varphi_2, G_{sub}) = \frac{1}{N} \sum_{i=1}^N Dense(G_i, G_{sub_i}; \varphi_2) - \log \frac{1}{N} \sum_{i=1, j \neq i}^N e^{Dense(G_i, G_{sub_j}; \varphi_2)} \quad (8)$$

where $Dense(\cdot)$ is the dense network of several MLP layers with the concatenation of the embedding of G and G_{sub} as the input.

To maximize Equation (7) and achieve subgraph information bottleneck, the lower bound of $I(Y, G_{sub})$ need to increase as much as possible and the upper bound of $I(G, G_{sub})$ needs to decrease. Therefore, we begin by performing gradient descent on the negative of \mathcal{L}_{MI-pro} within the dense network. The goal is to maximize \mathcal{L}_{MI-pro} and achieve the upper bound of $I(G, G_{sub})$, thus satisfying the constraints of Equation 11. After that, we utilize gradient descent to optimize Equation (11). By optimizing \mathcal{L}_{cls} , we increase the lower bound of $I(Y, G_{sub})$, and by optimizing \mathcal{L}_{MI-pro} , we decrease the upper bound of $I(G, G_{sub})$. This approach aims to maximum Equation (7) and seek the subgraph information bottleneck.

We summarized the details in section ‘‘Subgraph learning in ZeroBind for potential binding pockets in proteins’’ of the revised manuscript

7. Q: I also suggest authors use bold capitalized letters for the matrix and bold lowercase letters for the vector to discriminate it from the scalar.
A: Thanks for your suggestions. The letters in Equations have been standardized in the revised manuscript.

Response to Reviewer2

Key results

The paper presents a novel meta-learning framework for predicting protein-drug interactions, addressing the challenge of learning interactions involving new drugs or proteins. The core architecture of the proposed model is a graph neural network that represents both the drug and protein structures as graphs, incorporating atom features and pre-trained residue embeddings as node features. To capture the binding pocket of the protein, a subgraph information bottleneck technique is employed.

A: Thanks for your comments on our manuscript. Below comments are constructive to strengthen the manuscript. We have revised the manuscript according to the below comments.

1. Q: Figure 2b does not provide conclusive evidence regarding the correlation between the number of training proteins and the performance of the models (DeepConv, GraphDTA, DeepPurpose, AIbind, and Zerobind). The plot suggests that there may not be a clear relationship between these factors. However, further analysis and statistical tests are needed to draw definitive conclusions.

A: Thanks for your comments. If we understand correctly, you may ask the correlation between the number of the training molecules (not proteins) and the performance of the models. Figure 2b mainly shows AUROC comparison of the protein-specific ZeroBind and the baseline methods for 775 proteins. We have additionally provided Figure 2c, which presents the number of proteins that the method performs the best among the six methods for different ranges of the binding molecule numbers. We can see that ZeroBind outperforms other baseline methods for most proteins with only a few known binding drugs, followed by AI-Bind.

2. Q: The authors introduce the Jaccard similarity between the learned binding pocket and the true binding pocket. From Figures 2a and 2b, it appears that the false positive rate is relatively high. Changing the binding pocket can indeed alter the binding configuration, leading to different binding affinity or interaction. There is a concern that the Subgraph Information Bottleneck (SIB) module may learn factors other than the binding pocket since the entire process is unsupervised. To address this concern, it would be valuable to investigate the performance when the true binding pocket is provided or when the SIB module is trained in a supervised manner. Additionally, conducting an ablation study could further support the argument for the validity of the SIB module.

A: The binding pocket data is far less than the DTI data, and the protein structures predicted by AlphaFold2 still have no known binding pockets. Of the 1603 proteins in the benchmark dataset, only 535 proteins have part of known binding pockets. Thus, it is difficult to train the SIB module in ZeroBind for all proteins to detect binding pockets using the local binding pocket labels. Instead, we train SIB module of ZeroBind using the global DTI binding labels, which potentially guides the SIB module to locate the potential pockets in proteins. We discuss this point in the Discussion section of the revised manuscript.

In addition, we presented the ablation study about the SIB module and we further conduct an experiment by setting the node assignment matrix Z in Equation (4) randomly. The results show that randomly selected residues as binding pockets almost have no overlap with true binding pockets (Figure 3a and b), and the Jaccard similarity is much smaller than that of ZeroBind for predicted binding pockets. The results prove the validity of the SIB module for identifying potential binding pockets in an unsupervised way. The results are also shown in Ablation studies part. In summary, the results indicate that the SIB module in ZeroBind learns the potential binding pockets instead of other unrelated factors.

3. Q: The paper highlights three primary contributions: meta-learning, task adaptive self-attention, and IB-subgraph. However, it should be noted that these methods have been previously employed in the drug-protein prediction task, so the level of novelty may not be particularly high. For instance, MetaDTA has utilized similar techniques for learning and task adaptive self-attention. Additionally, the idea of learning the binding pocket unsupervised through IB-

subgraph has been extensively explored, as seen in studies like GEFA. To demonstrate the effectiveness of the proposed approach, the authors should compare their results with these baseline methods and clearly showcase the improvements achieved through their design.

A: Thanks for your comments. We mentioned both MetaDTA and GEFA in the Introduction section of the revised manuscript.

MetaDTA is a ligand-based DTI prediction method under the meta-learning framework, it is a few-shot predictor for proteins with a few known binding drugs. MetaDTA uses Multi-head Cross-Attention network to capture the relationship between the support and the query ligands instead of different proteins. In contrast, ZeroBind utilizes a Task Adaptive Self-Attention module to learn the importance of different tasks of proteins. MetaDTA does not use any protein information and cannot support zero-shot prediction for unseen proteins without any known binding drugs in the training set. Our method ZeroBind is focused on the zero-shot prediction, although ZeroBind supports few-shot prediction, and we also demonstrate the effectiveness of ZeroBind for few-shot prediction. In addition, MetaDTA is still an ICLR conference workshop paper, and did not provide the source codes publically for independent running. Thus, it is difficult for us to compare our method with MetaDTA.

GEFA combines pre-trained protein embeddings and a graph-in-graph neural network with attention mechanism to capture the interactions between drugs and protein residues, it does not use any subgraph-based methods, including IB-subgraph, to learn potential binding pockets. We also searched the literature and did not find any published methods that uses subgraph matching to identify potential binding pockets in the proteins.

GEFA requires 2D contact map information, embeddings, secondary structure element and solvent accessibility as input features, which require third-party tools to obtain them. We tried to run their source codes on our benchmark datasets, but it failed since the authors of GEFA did not provide Readme (<https://github.com/ngminhtri0394/GEFA>) about how to run the method on new datasets and make it difficult to run it on our datasets. We opened an issue at GEFA Github, but we still did not receive a response.

To validate the effectiveness of our method, we added another recently published baseline method DrugBAN² to compare with our method and the results show that ZeroBind is superior to DrugBAN.

4. Q: It is important for the authors to include details about the data used in the COVID drug section. Without this information, it becomes difficult to verify the results of the experiment conducted in that particular domain.

A: The PDB IDs for protein structures used in the COVID drug section are shown in Supplementary Table 3 and the data is also available at <http://www.csbio.sjtu.edu.cn/bioinf/ZeroBind/datasets.html> and the Github repository <https://github.com/myprecioushh/ZeroBind/tree/main/data>.

References

- 1 Xie, Z. W. & Xu, J. B. Deep graph learning of inter-protein contacts. *Bioinformatics* **38**, 947–953, doi:10.1093/bioinformatics/btab761 (2022).

- 2 Bai, P., Miljković, F., John, B. & Lu, H. Interpretable bilinear attention network with domain adaptation improves drug-target prediction. *Nature Machine Intelligence* **5**, 126-136, doi:10.1038/s42256-022-00605-1 (2023).

REVIEWERS' COMMENTS

Reviewer #2 (Remarks to the Author):

The authors have adequately addressed the majority of my concerns. However, I still have one remaining question:

In relation to the ablation study concerning the SIB module for the binding pocket, I find that randomly selecting residues as the binding pocket represents a rather simplistic baseline. This approach does not convincingly demonstrate whether SIB can effectively learn the binding pocket. To further investigate this aspect, I would suggest conducting an experiment using a subset of 535 proteins that have well-defined binding pockets. In this experiment, the Z component can be replaced with the ground truth binding pocket. If SIB is indeed capable of learning the binding pocket, the performance difference between the original experiment and this modified version should not be too significant.

Reviewer #3 (Remarks to the Author):

The manuscript didn't show enough novelty in comparison to previous works, and as a result the improvement is quite marginal relative to graphDTA, AI-bind, etc. As for the prestigious journal, I would like to see wet experimental validation .

We carefully addressed all the comments. The changes have been highlighted in red in the revised manuscript. The detailed responses to the reviewers are attached below and highlighted in blue. We highly appreciate the constructive comments, which are very helpful in strengthening this paper.

Response to reviewer 2

REVIEWERS' COMMENTS

Reviewer #2 (Remarks to the Author):

The authors have adequately addressed the majority of my concerns. However, I still have one remaining question:

In relation to the ablation study concerning the SIB module for the binding pocket, I find that randomly selecting residues as the binding pocket represents a rather simplistic baseline. This approach does not convincingly demonstrate whether SIB can effectively learn the binding pocket. To further investigate this aspect, I would suggest conducting an experiment using a subset of 535 proteins that have well-defined binding pockets. In this experiment, the Z component can be replaced with the ground truth binding pocket. If SIB is indeed capable of learning the binding pocket, the performance difference between the original experiment and this modified version should not be too significant.

R: Thanks for your comments. As suggested, we train a variant ZeroBind of using true binding pocket as the node assignment matrix Z for the inductive test set. This variant ZeroBind achieves an average AUROC of 0.8278, which is higher than the AUROC 0.8032 of the ZeroBind that uses learned node assignment matrix Z by a narrow margin. The experimental results further validate the effectiveness of the IB-subgraph module in ZeroBind. We mentioned this result in third paragraph of the section "ZeroBind detects the subgraphs that align well with known binding pockets of proteins in a weakly supervised way"

Response to reviewer 3

Reviewer #3 (Remarks to the Author):

The manuscript didn't show enough novelty in comparison to previous works, and as a result the improvement is quite marginal relative to graphDTA, AI-bind, etc. As for the prestigious journal, I would like to see wet experimental validation .

R: Thanks for your comments. As mentioned in our manuscript, we summarized our contribution as follows:

1) We propose model-agnostic IB-subgraph learning to automatically discover compressed subgraphs as potential binding pockets in proteins instead of redundant graph information derived from the whole protein. To date, there are still no existing methods that using subgraph to detect potential binding pockets in proteins. graphDTA, AI-bind only can be used to predict DTIs without too much biological interpretability.

2) We conduct extensive experiments on three independent zero-shot test sets and one few-shot test set. Results show that ZeroBind consistently outperforms existing methods. Compared to the second best method, ZeroBind achieves relative improvements in AUROC with 2.86% on the Transductive test set, 10.29% on the Semi-inductive test set, and 3.38% on the Inductive test set, the corresponding t-test p-value is 1.02×10^{-6} , 8.02×10^{-11} , 3.06×10^{-7} , respectively. We can see that ZeroBind significantly outperforms existing methods.

3) We are a pure computational biology group, it is very difficult for us to validate the results using wet-lab experiments. Further validation of real-world SARS-COV-2 drug-target binding prediction demonstrates the reliability of ZeroBind predictions using docking software, which is widely used to evaluate the detected drug-target interactions.